biomimetics/fluid mechanics/ocean engineering

leading-edge tubercles, ducted propeller, underwater radiated noise, FW-H, CFD

**Author for correspondence:**
Weichao Shi
e-mail: weichao.shi@strath.ac.uk

# Hydroacoustic and hydrodynamic investigation of bio-inspired leading-edge tubercles on marine-ducted thrusters

## Callum Stark and Weichao Shi

Naval Architecture, Ocean and Marine Engineering, University of Strathclyde, Glasgow, UK

CS, 0000-0001-7391-5197; WS, 0000-0001-9730-7313

Underwater radiated noise (URN) has a negative impact on the marine acoustic environment where it can disrupt marine creature's basic living functions such as navigation and communication. To control the ambient ocean noise levels due to human activities, international governing bodies such as the International Maritime Organization (IMO) have issued non-mandatory guidelines to address this issue. Under such framework, the hydroacoustic performance of marine vehicles has become a critical factor to be evaluated and controlled throughout the vehicles' service life in order to mitigate the URN level and the role humankind plays in the ocean. This study aims to apply leading-edge (LE) tubercles of the humpback whales' pectoral fins to a benchmark ducted propeller to investigate its potential in noise mitigation. This was conducted using CFD, where the high-fidelity improved delayed detached eddy simulations (IDDES) in combination with the porous Ffowcs-Williams Hawkings (FW-H) acoustic analogy was used to solve the hydrodynamic flow field and propagate the generated noise to the far-field. It has been found that the LE tubercles have shown promising noise mitigation capabilities in the far-field, where the OASPL at $J = 0.1$ was reduced to a maximum of 3.4 dB with a maximum of 11 dB reduction in certain frequency ranges at other operating conditions. Based on detailed flow analysis researching the fundamental vortex dynamics, this noise reduction is shown to be due to the disruption of the coherent turbulent wake structure in the propeller slipstream causing the acceleration in the dissipation of turbulence and vorticity-induced noise.

# 1. Introduction

Noise pollution is of growing concern with well documented negative effects on marine lives where the acoustic environment is vital to perform key functions such as communicating, finding prey and navigation. The shipping industry contributes significantly to the noise pollution in the world's oceans. Due to the awareness of such detrimental environmental impacts being raised, in 2014, the International Maritime Organisation (IMO) set non-mandatory guidelines in order to reduce/limit the underwater radiated noise (URN) from commercial shipping [1]. More specifically, the propulsion system is generally the largest proportion of the URN and thus, reducing the radiation from this source would provide major benefits to the ocean community. While mitigating URN is a pressing goal among international bodies, it is also equally important to maintain its functionality of being a propulsion unit providing sufficient thrust in an energy-efficient manner. Therefore, the technological solutions for noise mitigation shall not be at the price of compromising the propulsive efficiency.

To develop the next generation noise mitigation propulsion solutions, an accurate URN prediction method is needed to support the design and optimization. Numerical prediction methods have become the frontline of research for URN. With the growing computational power, predicting the URN in the early design stage of marine vessels is now becoming a routine practice. The most common methodology to predict URN is by using a hybrid method, whereby an incompressible solver is used to solve the hydrodynamic flow field and then an acoustic analogy is used to simulate the noise propagation to the far-field. The most popular acoustic analogy for hydroacoustics is the Ffowcs-Williams Hawkings (FW-H) method. Various studies have been conducted using different formulations of the FW-H acoustic analogy in both cavitating and non-cavitating flows [2–7]).

The hybrid method has also contributed to developing noise reduction solutions for marine propellers, particularly for ducted propellers [2,8,9]. Qin *et al.* [8] and Sun *et al.* [9] used detached eddy simulation (DES) and the FW-H analogy to predict the noise reduction capability of the pumpjet propulsors with trailing edge serrations on decelerating ducts, which has demonstrated noise reduction but with a compromised hydrodynamic performance. It is found that the noise reduction is due to the disruption of the coherent vortex structure in the propeller slipstream, breaking down big vortex structures into smaller ones which can then be quickly attenuated to heat. Although this concept is promising, applying noise mitigation technologies to the decelerating duct restricts its use to military applications as they are not widely used within the shipping industry.

On the other hand, most other noise mitigation strategies are focused on the cavitating propeller, such as pressure pore technology and the roughness application to unload the propeller tip and reduce tip vortex cavitation [10,11]. This does come at the cost of hydrodynamic performance, therefore highlighting the challenge within the maritime industry, maintaining hydrodynamic performance while simultaneously reducing the URN or vice-versa.

As most noise mitigation strategies either focused on open propellers in cavitating conditions, less attention is given to ducts that are used in a more commercial setting and how they can be modified to reduce URN. Reducing cavitation and its associated URN is crucial as it is the most dominant noise source, but in non-cavitating conditions, reducing the noise signature associated with turbulent and vorticity mechanisms becomes more important [2]. Even more so, a duct such as the accelerating 19A duct [12] is a more widely used duct as it is used on a number of different craft, such as tug boats, fishing vessels and underwater vehicles, due to its high thrust capability in the heavy-loaded conditions, compared to the military restricted decelerating duct which is used to reduce cavitation and its subsequent URN [13]. Therefore, focusing on reducing the URN signature from a ducted propeller that is widely used within the maritime community will have a larger positive impact on the ocean environment. In order to achieve this, one can look to mother nature for inspiration as humankind has done so many times in order to answer our problems.

Ironically, a marine mammal that is negatively impacted by the by-products of marine vessels is the inspiration behind this biomimetic research. The humpback whale is a magnificent creature that despite its large and cumbersome build can catch prey using acrobatic, agile manoeuvres. Humpback whales also have small bumps on the pectoral fins known as leading-edge (LE) tubercles, which give the fin a scalloped appearance [14]. Fish & Battle [15] first proposed that the morphology and placement of LE tubercles would suggest that they acted as enhanced lift devices to control flow over the flipper and maintain lift at high angles of attack while also improving the manoeuvrability associated with the whale's unique feeding behaviour.

LE tubercle's effect on aero/hydrodynamic performance has been well researched, showing benefits for industrial applications such as aerofoils, fans, propellers, control surfaces, wind and tidal turbines [16–24]. The fundamental fluid dynamics of LE tubercles has also been investigated. Wei *et al*. [25] conducted an experimental investigation into the flow separation control behaviour of LE tubercles on hydrofoils, showing that counter-rotating vortex pairs are generated over each tubercle which can significantly modify the boundary layer flow through chordwise and spanwise interactions. Wei *et al*. [26] conducted an experimental investigation into the influence of LE tubercles on the aerodynamic performance of tapered swept-back wings. Through flow visualization techniques, it was shown that LE tubercles produce a pattern of nodes and saddles in the surface flow that is significantly tilted due to the presence of the sweep angle. Troll *et al*. [27] conducted high-fidelity computational simulations to show that such counter-rotating vortex pairs from tubercles can interfere and disrupt the main tip vortex structure, encouraging the dispersion of the tip vortex when compared to the reference foil. Similarly, the inclusion of LE tubercles and their influence on noise has been investigated as this is key to many industries including the maritime industry.

Hansen *et al*. [28] investigated the reduction of aerofoil tonal noise with the inclusion of LE tubercles in a wind tunnel at a Reynolds number of 120 000. It was concluded that the implementation of the bio-inspired concept produced a reduction in noise, where the largest amplitude and the smallest wavelength configuration provided the greatest reduction in tonal and broadband noise. This is believed to be due to the disruption of the coherent trailing edge wake structure by the counter-rotating vortices generated by the LE tubercles. Wang *et al*. [29] conducted large eddy simulations (LES) on a popular NACA foil, and it was shown that a reduction of 13.1 dB-13.9 dB was possible with the inclusion of the LE tubercles. Lau *et al*. [30] investigated the effect of LE tubercles on aerofoil gust-interaction noise. It was shown that aerofoil gust-interaction noise could be reduced by up to 80% with the inclusion of LE tubercles. Clair *et al*. [31] confirmed the noise reduction effect of LE tubercles based on a NACA 651210 aerofoil in low-velocity flows both numerically and experimentally. Narayanan *et al*. [32] conducted an experimental investigation into LE serrated flat plates and foils and their ability to reduce the broadband noise generated due to the interaction between the foil LE and the impinging turbulence. It was concluded that the LE serrated flat plate and aerofoil could reduce the noise by a maximum of 9 and 7 dB, respectively, when compared with their baseline counterpart. Kim *et al*. [33] conducted numerical simulations to investigate the LE tubercle's ability to reduce the aerofoil-turbulence interaction noise, showing that the overall sound pressure level (OASPL) decreased monotonically (linearly) with the amplitude of the LE tubercle-modified foil. Chaitanya *et al*. [34] conducted a detailed experimental investigation into the effectiveness of sinusoidal LE serrations on aerofoils for the reduction of the noise generated by the interaction with the turbulent flow; it was concluded that trailing edge noise could be reduced by up to 3 dB. Turner & Kim [35] used high-accuracy numerical simulations to conduct an aeroacoustic study into LE tubercle-modified aerofoils and to investigate their source mechanisms. It was deemed that the exponential growth of noise reduction with frequency, reported by [32–34], is unexplained solely by the source mechanisms. Zhang & Frendi [36] used numerical simulations to investigate the influence of LE tubercles on the noise from an aerofoil in the wake of a cylinder, showing that the near-field sound pressure level was found to be between 4 and 10 dB lower for the wavy LE aerofoil depending on location. Tong *et al*. [37] used LES to show that LE tubercle-modified aerofoil could reduce the far-field noise by an average of 9.5 dB at the azimuthal angle of 90° when compared to the reference design. In maritime applications, there is a lot of potential to implement LE tubercles for noise reduction purposes. Shi *et al*. [22,23] conducted experimental trials of LE tubercles on tidal turbines, showing cavitation can be constrained and hence the radiated noise can be reduced; more recently, a similar cavitation fencing pattern has been observed on tubercle-modified propellers by [24]. In summary, LE tubercles have shown promising noise mitigation capabilities in aviation and maritime applications; however, their hydroacoustic benefits on marine propulsors have not been an area of focus in the current state of the art.

More recently, tubercles have been applied to the ducts of ducted propellers. Stark *et al*. [38] initiated a preliminary design optimization study for LE tubercles as applied to the duct of a benchmark ducted propeller using both actuator disc and propeller resolved methods coupled with a RANS-based method. It was found that LE tubercles could improve the maximum duct thrust capability of the ducted propulsor; however, this was dependent on the wavelength and amplitude of the tubercle. At the maximum efficiency point, flow separation compartmentalization was observed; however, this did not result in a duct thrust improvement using a RANS-based method. However, it was recommended to use a scale-resolved method such as DES to further investigate the performance of the tubercle-ducted propeller in flow-separating conditions [39].

To the author's knowledge, LE tubercles as applied to the ducted propeller have not been investigated for potential hydroacoustic benefits, despite the various promising evidence on other applications as previously discussed. Due to the ducts' aerofoil-like cross-section and strong evidence suggesting LE tubercles can mitigate noise generation; this investigation has been initiated. This study aims to use a hybrid approach, implementing a high-fidelity DES solver with the FW-H acoustic analogy to establish the noise mitigation capabilities of this well-researched concept on a novel application in addition to potential hydrodynamic benefits by in-depth analysis of key performance variables and wake vortex dynamics.

# 2. Numerical approach

## 2.1. Hydrodynamic solution

Commercial code STAR CCM+ was used to complete this study. Implicit unsteady incompressible DES was used to solve the hydrodynamic flow field and the rigid body motion (RBM) method was used to model the propeller revolutions. The specific formulation of DES used was the improved delayed detached eddy simulation (IDDES); this solves the near-wall regions using a RANS approach, in this case using the SST k-omega turbulence model, with the LES solver used to solve the rest of the domain. DES was used as opposed to RANS as it has been shown that the RANS solver does not capture the instabilities within the propeller wake flow [40], and in the case of tubercles, the suitability of the RANS solver has been questioned, especially in flow separation conditions [41]. In addition, it has been recommended to use high-fidelity solvers such as DES/LES when predicting the URN of marine propulsors [7,42]. The rotation rate was kept constant at 15rps and the advance velocity was varied to predict the open-water curve characteristics. The Reynolds number was estimated as $1.05 \times 10^6$ using the blade rotation rate.

RBM was selected as it is considered the most accurate methodology to predict unsteady propeller flow [43], where a time-step of 1 degree of rotation per time-step was selected which is roughly $1.85 \times 10^{-4}$ s, a time-step of between 0.5 and 2 degrees is recommended by ITTC [44]. The second-order scheme was applied to the temporal discretization. The ducted propeller ran for 10 revolutions using the RBM method, where hydrodynamic variables had converged, and the propeller wake flow had developed sufficiently downstream. The operating conditions where the acoustic data were collected were run for an additional 10 propeller revolutions to allow a period to collect the pressure fluctuations. The acoustic analogy was validated by comparing hydrodynamic and hydroacoustic pressures at a near-field receiver located at a radial position from the open propeller centre used within the investigation.

## 2.2. Hydroacoustic solution

The FW-H formulation was used to propagate the sound to the far-field [45]. The governing equation to describe the FW-H formulation can be shown in equation (2.1).

$$\left(\frac{1}{c_0^2}\frac{\partial^2}{\partial t^2} - \nabla^2\right)\hat{p}\,(x,t) = \frac{\partial}{\partial t}\{[\,\rho_0 v_n + \rho(u_n - v_n)]\delta(f)\} - \frac{\partial}{\partial x_i}\{[\Delta P_{ij} + \rho u_i(u_n - v_n)]\delta(f)\} + \frac{\partial^2}{\partial x_i \partial x_j}\{T_{ij}H(f)\}T_{ij},$$
(2.1)

$$T_{ij} = \rho u_i u_j + P_{ij} - c_0^2\tilde{\rho}\delta_{ij},$$
(2.2)

and
$$\tilde{\rho} = \rho - \rho_0,$$
(2.3)

where $c_0$ denotes the sound velocity in the far-field and $\hat{p}$ is sound pressure at the far-field ($\hat{p} = p - p_0$). $u_i$ and $v_i$ are fluid and surface velocity components, respectively. $n$ indicates the projection along the outward normal to the surface. Lighthill Stress tensor can be written as equation (2.2). $\tilde{\rho}$ is the density perturbation of the fluid where $\rho$ denotes the density of the fluid and $\rho_0$ is the free stream density as defined in equation (2.3).

The FW-H acoustic analogy considers all fundamental noise sources—monopole (thickness of blades and cavities), dipole (blade loading) and quadrupole (nonlinear contributions associated with turbulent structures, products of cavity destruction and vortex–vortex interaction in the propeller slipstream). While the contributions from monopole and dipole sources are evaluated by surface integrals on the respective noise sources, for example the propeller blade, the quadrupole term requires volume integration, which is very expensive. This is known as the impermeable approach, which generally focusses on the linear terms

as calculating the volume integral quadrupole term is very costly. But, as has been shown, nonlinear terms associated with propeller noise cannot be neglected in the far-field [4,5]. Therefore, the porous surface approach was developed, a formulation derived by [46] in which the surface surrounds the propeller and as much of a portion of the turbulent wake structure as possible. In such formulations, for example Farassat Formulation 1A [47], the terms responsible for monopole and dipole contributions lose their original meaning and become pseudo-monopole and pseudo-dipole terms, which include also the contribution from the quadrupole (nonlinear) term, as long as the permeable control surface encompasses a portion of the turbulent wake. This results in all noise sources being solved by a surface integral, which includes the quadrupole term while negating the need for a costly volume integral computation. The equation for the FW-H permeable surface approach and an incompressible flow assumption can be defined firstly by the two acoustic variables in equations (2.4) and (2.5).

$$U_i = u_i \tag{2.4}$$

and

$$L_i = P_{ij}\widehat{n}_j + \rho u_i(u_n - v_n), \tag{2.5}$$

where $u$ and $v$ are the fluid and porous surface velocities. In the assumption of incompressible flow and a fixed, stationary data surface, $\rho = \rho_0$ and $v_n = 0$, the permeable surface FW-H equation simplifies to the definition in equation (2.6).

$$4\pi p(x,t) = \int_S \left[\frac{\rho_0 \dot{U}_n}{r}\right]_\tau dS + \int_S \left[\frac{\dot{L}_r}{c_0 r}\right]_\tau dS \int_S \left[\frac{L_r}{r^2}\right]_\tau dS, \tag{2.6}$$

where $r$ is the radiation direction and the dot defines a source time derivative with respect to retarded time. Subscripts $r$ and $n$ define the product of a quantity with a unit vector either radiation or normal directions, respectively.

Acoustic pressure is collected in the time domain at each time-step. By using Fast Fourier Transform (FFT), it is transferred to the frequency domain and then SPL values are calculated in the frequency domain as follows in equation (2.7);

$$\mathrm{SPL} = 20\log\left(\frac{p}{p_{\mathrm{ref}}}\right), \tag{2.7}$$

where $p$ is acoustic pressure in the frequency domain, Pa; $p_{\mathrm{ref}}$ (Pa) is reference acoustic pressure (for water $p_{\mathrm{ref}} = 1 \times 10^{-6}$ Pa). In addition, OASPL is calculated by equation (2.8):

$$\mathrm{OASPL} = 20\log\left(\frac{P_{\mathrm{rss}}}{P_{\mathrm{ref}}}\right), \tag{2.8}$$

where $P_{\mathrm{rss}}$(Pa) is total acoustic pressure, which is obtained within this study by summing up all acoustic pressures in the third-octave band frequency domain in accordance with the root sum square rule.

# 3. Test case

## 3.1. Overview

The reference geometry 'REF' was selected as the benchmark 19A-ducted propeller and Kaplan series, KA4–55 propeller; detailed geometry can be found in [12]. The rendered geometry can be found in figure 1a and the parameters in table 1.

The tubercle duct was created as an idealized sinusoidal waveform with an amplitude of 5 mm ($A/c = 0.05$) and a tubercle count of 10 ($\lambda/c = 0.75$), labelled as 'SLE' duct. The three-dimensional design can be shown in figure 1b. The geometrical parameters of the tubercle were selected based on a previous optimization study where several amplitude and wavelength configurations were investigated. It was noted that an increase in amplitude and reduction in wavelength degraded the hydrodynamic performance of the ducted propeller in the lower $J$ range. However, the current design configuration produced an improvement in duct thrust capability at the heavy-loaded condition, while showing compartmentalization of flow separation at the higher $J$ range where flow separation occurs and was therefore selected for further analysis [38].

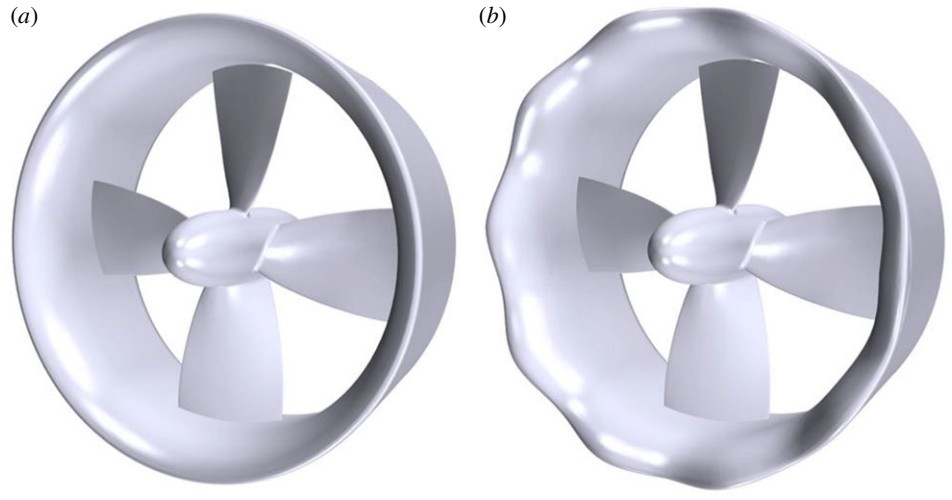

**Figure 1.** Ducted propeller geometry. (*a*) REF and (*b*) SLE duct.

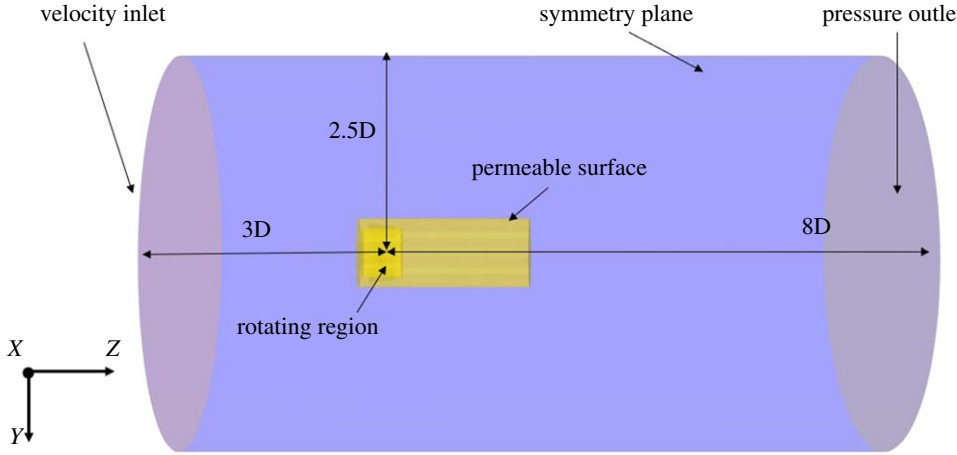

**Figure 2.** Computational domain (distances taken from propeller centre).

**Table 1.** Geometrical parameters of reference ducted propeller.

| variable (duct) | unit | variable (propeller) | unit |
|---|---|---|---|
| Type | 19A | type | Kaplan |
| outer diameter, Dd | 0.306 m | blade number | 4 |
| inner diameter, Di | 0.254 m | expanded area ratio (EAR) | 0.55 |
| chord, $L_{DUCT}$ | 0.125 m | pitch-diameter ratio (P/D) | 1 |
| | | diameter, D | 0.25 m |
| | | tip clearance, t | 2 mm |
| | | position wrt duct | $0.5L_{DUCT}$ |

## 3.2. Computational domain

The computational domain consisted of a cylindrical domain, where the propeller was located 3D from the inlet and 8D from the outlet and 2.5D from the outer circumferential wall. The inlet was defined as a velocity inlet, outlet as pressure outlet and symmetry plane on the circumferential face as shown in figure 2. The duct and propeller were defined as no-slip wall boundaries.

(*a*)    (*b*)

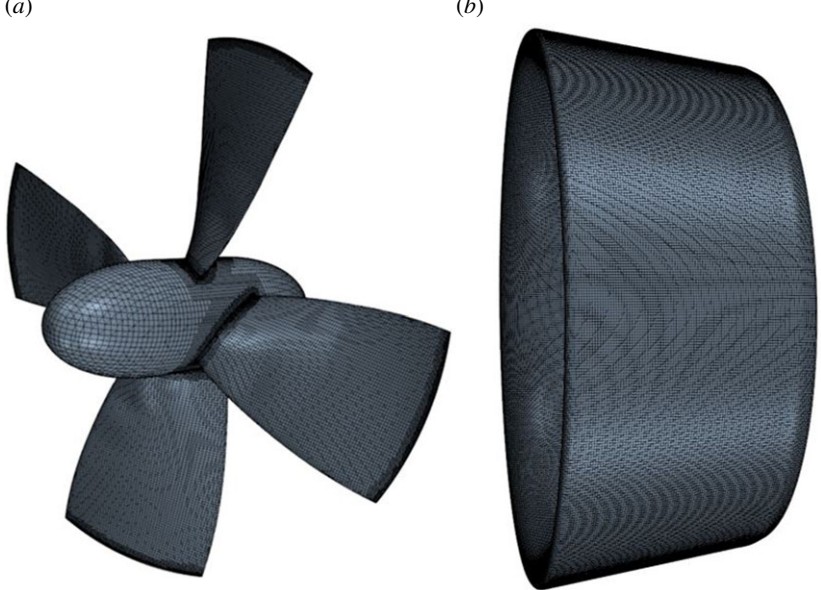

**Figure 3.** Surface mesh of the propeller blade and duct. (*a*) Blade mesh and (*b*) duct mesh.

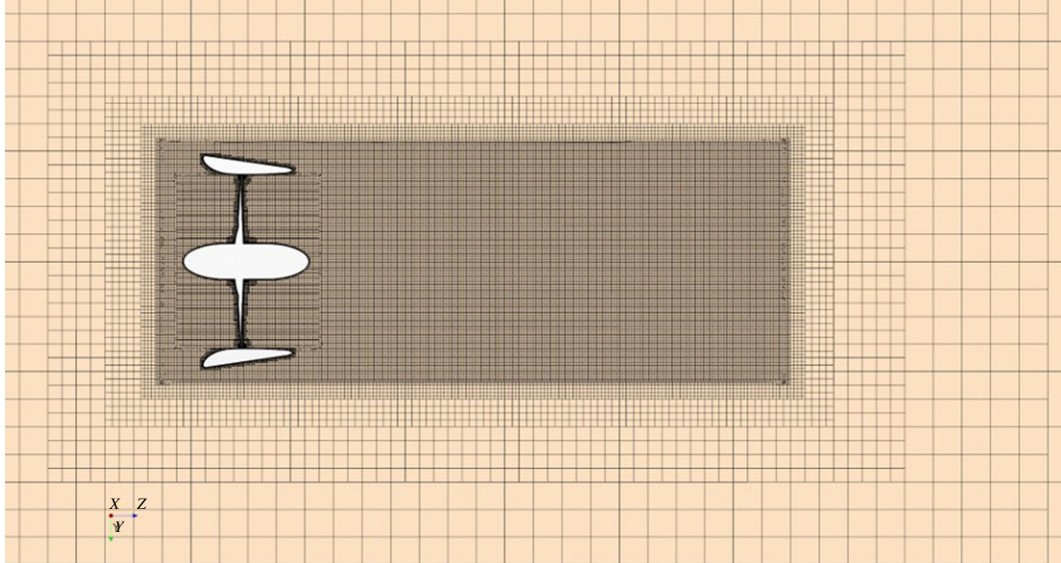

**Figure 4.** Volume mesh of computational domain.

## 3.3. Mesh generation

The mesh was generated using unstructured hexahedral mesh to the count of approximately 13 million cells, where prism layers were used to resolve the boundary layer. A low $y^+$ wall approach was employed, where the average $y^+ < 1$, with a maximum of roughly $y^+$ of 2.3 located on the blade LE. An interface prism layer between the rotating region of the propeller and the surrounding static region was created to ensure mesh alignment between regions and reduce numerical diffusion across the interface. A volumetric control was selected to align with the permeable region, maintaining a uniform mesh and reducing the level of numerical diffusion due to change in mesh cell size in the permeable region [6]. Further volumetric controls were applied to allow a smooth transition to the core mesh. The blade and duct surface mesh can be shown in figure 3 and a section of volume mesh is shown in figure 4.

## 3.4. Permeable surface design and receiver location

The permeable surface was 0.7D from the propeller plane and 3.5D in length to allow for a portion of the turbulent wake structure to be accounted for within the noise prediction, as it has been stipulated that the

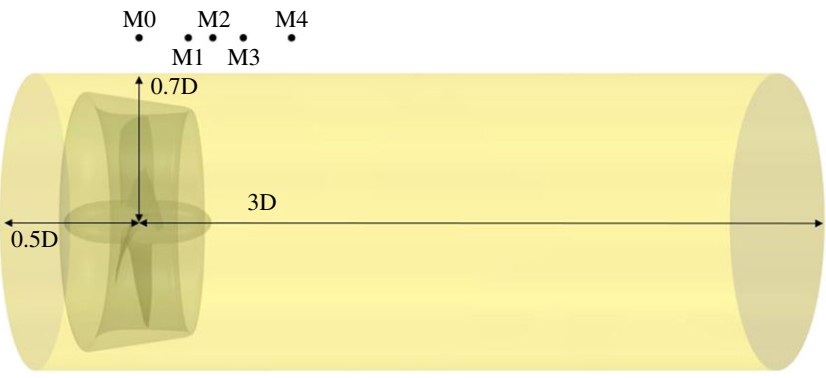

**Figure 5.** Permeable surface (end-caps removed) and near-field receiver locations.

**Table 2.** Near-field receiver locations.

| receiver | location (m) |
| --- | --- |
| M0 | [0.0, 0.25, 0.0] |
| M1 | [0.0, 0.25, 0.1] |
| M2 | [0.0, 0.25, 0.15] |
| M3 | [0.0, 0.25, 0.2] |
| M4 | [0.0, 0.25, 0.3] |

quadrupole term—mainly dominated by turbulence and vorticity-induced noise—cannot be neglected in the far-field noise analysis of marine propellers [5]. Additionally, the inclusion of tubercles has shown to have an influence on the vortical wake structure of the duct, and to observe this in the noise prediction, the inclusion of the nonlinear noise source is paramount. It is also recommended to remove the 'end-caps' from the permeable surface due to the wake structure crossing over the region, resulting in contamination of the noise signature, and this was done for this study [6]. In terms of permeable surface dimensions, the exact dimensions and location of the integral surface have not been defined clearly in the present literature, and it is still being investigated [7]. The permeable surface and the near-field receiver positions can be shown in figure 5, and the location of the near-field receivers in terms of x, y and z is tabulated in table 2. The near-field receivers were used to validate the FW-H analogy against the direct hydrodynamic pressures. The near-field locations can be shown in figure 5. Far-field receivers were located at 100D from the propeller, and at increments of 30 degrees to cover the 360° range, which can be shown in the schematic in figure 6.

# 4. Verification and validation

## 4.1. Performance coefficients

The hydrodynamic performance of the ducted propeller was predicted using the traditional open-water characteristics. Ducted propeller performance coefficients can be outlined in equations (4.1.)–(4.6).

$$K_{TP} = \frac{T_P}{\rho n^2 D^4} \; , \tag{4.1}$$

$$K_{TD} = \frac{T_D}{\rho n^2 D^4}, \tag{4.2}$$

$$K_{TT} = K_{TP} + K_D, \tag{4.3}$$

$$K_Q = \frac{Q}{\rho n^2 D^5}, \tag{4.4}$$

$$ETA = \frac{K_{TT}J}{2\pi K_Q} \tag{4.5}$$

and

$$J = \frac{V_A}{nD}, \tag{4.6}$$

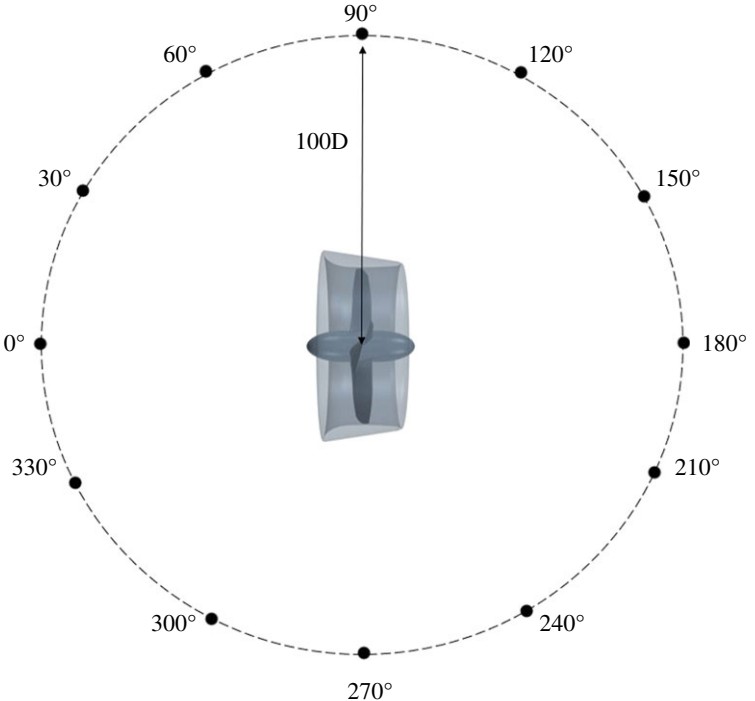

**Figure 6.** Location of far-field receivers.

**Table 3.** Uncertainty result for $K_{TT}$ and 10 $K_Q$.

|          | $\varphi_1$ | $\varphi_2$ | $\varphi_3$ | $R$  | %GCI$_{FINE}$ |
|----------|-------------|-------------|-------------|------|---------------|
| $K_{TT}$ | 0.178       | 0.177       | 0.172       | 0.13 | 0.23          |
| 10$K_Q$  | 0.279       | 0.278       | 0.276       | 0.33 | 0.83          |

where $K_{TP}$ is propeller thrust coefficient, $K_{TD}$ is duct thrust coefficient, $K_{TT}$ is total duct thrust coefficient, 10 $K_Q$ is the torque coefficient and ETA is open-water efficiency. Advance ratio, $J$, is defined by advance velocity $Va$ (m s$^{-1}$), $n$, rotation rate (rps) and propeller diameter D (m) and can be shown in equation (4.6). $T_P$ and $T_D$ are propeller and duct thrust (N) respectively, $Q$ is propeller torque (Nm) and $\rho$ is density (kg m$^{-3}$).

## 4.2. Mesh convergence

A verification study was conducted to determine the uncertainty of the numerical simulations. This was completed using the grid convergence (GCI) method first proposed by [48] and based on [49]. Additionally, this method is also recommended in the ITTC procedure [50]. The full methodology implemented in this study was defined by [51] and can be found within. The total thrust and torque coefficient were selected as the integral variable at advance ratio, $J = 0.55$, the operating condition at which maximum efficiency is achieved. The tabulated results can be shown in table 3.

The difference between the solution scalars ($\varepsilon$) should be determined by equation (4.7).

$$\varepsilon_{21} = \varphi_2 - \varphi_1, \quad \varepsilon_{32} = \varphi_3 - \varphi_2, \tag{4.7}$$

where $\varphi_1$, $\varphi_2$ and $\varphi_3$ represent the results using fine, medium and coarse mesh grids, respectively. The ratio of solution scalars is used to calculate the convergence condition by equation (4.8).

$$R = \frac{\varepsilon_{21}}{\varepsilon_{32}}. \tag{4.8}$$

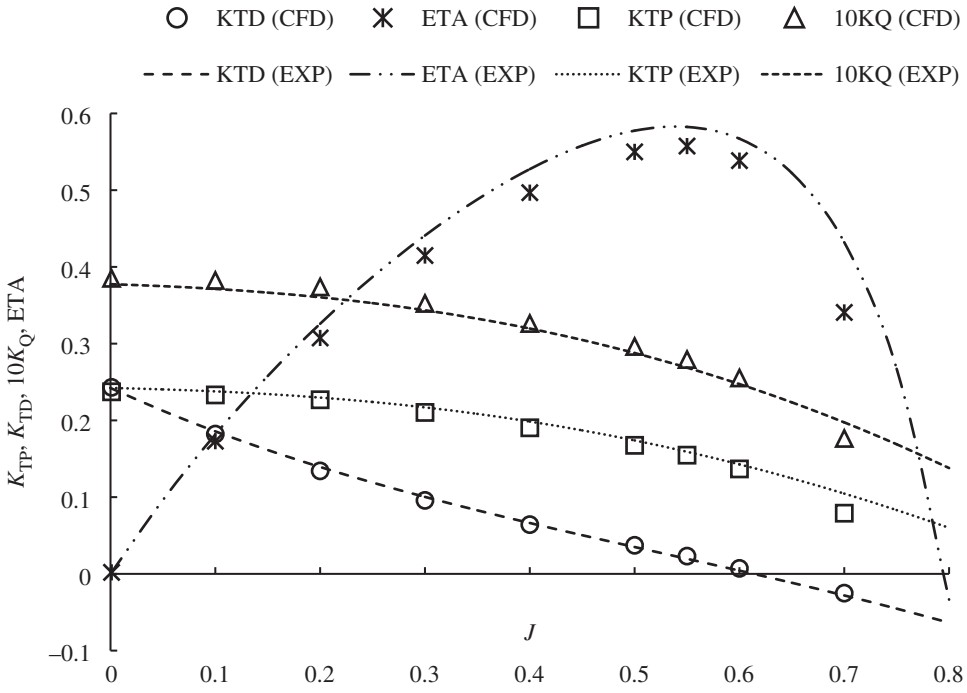

**Figure 7.** KA4–55 + 19A duct validation with experimental test.

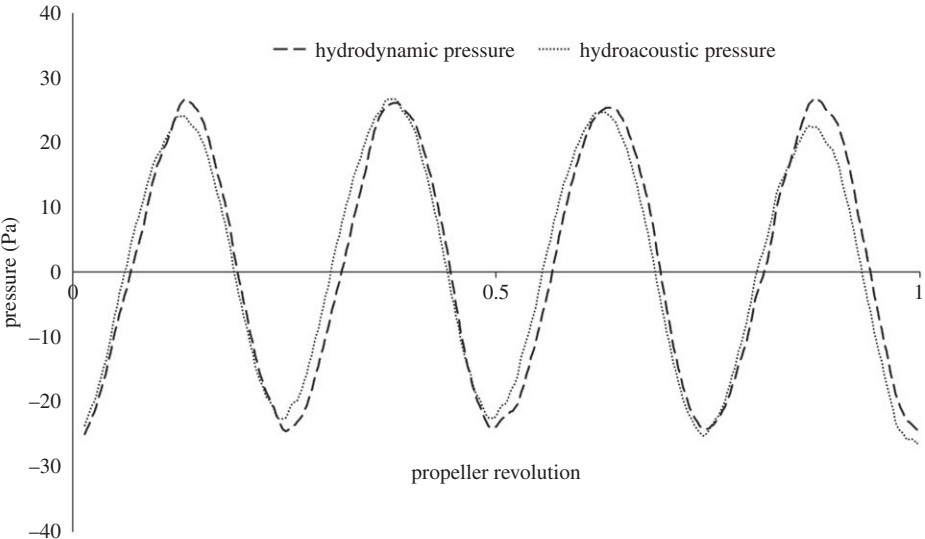

**Figure 8.** Near-field hydrodynamic and hydroacoustic pressure for open KA4–55 propeller at one propeller diameter from propeller centre (M0).

Solution type is determined with respect to the convergence condition, $R$: 1. oscillatory convergence, $-1 < R < 0$; 2. monotonic convergence $0 < R < 1$; 3. oscillatory divergence $R < -1$ and 4. monotonic divergence, $R > 1$. If $R$ is found as in case 2, the procedure can be directly employed. $GCI$ index is calculated by the following in equation (4.9):

$$\text{GCI}_{\text{fine}}^{21} = \frac{1.25 e_a^{21}}{r_{21}^p - 1},$$

(4.9)

where $p$ is apparent order and $e_a$ is an approximate relative error. Detailed information about the verification procedure can be found in [51]. Results obtained for the total thrust and torque coefficient and uncertainty level are given in table 3. As shown in table 3, the convergence condition (R) was between 0 and 1 (monotonic convergence). As a result of the uncertainty study at $J = 0.55$, the fine

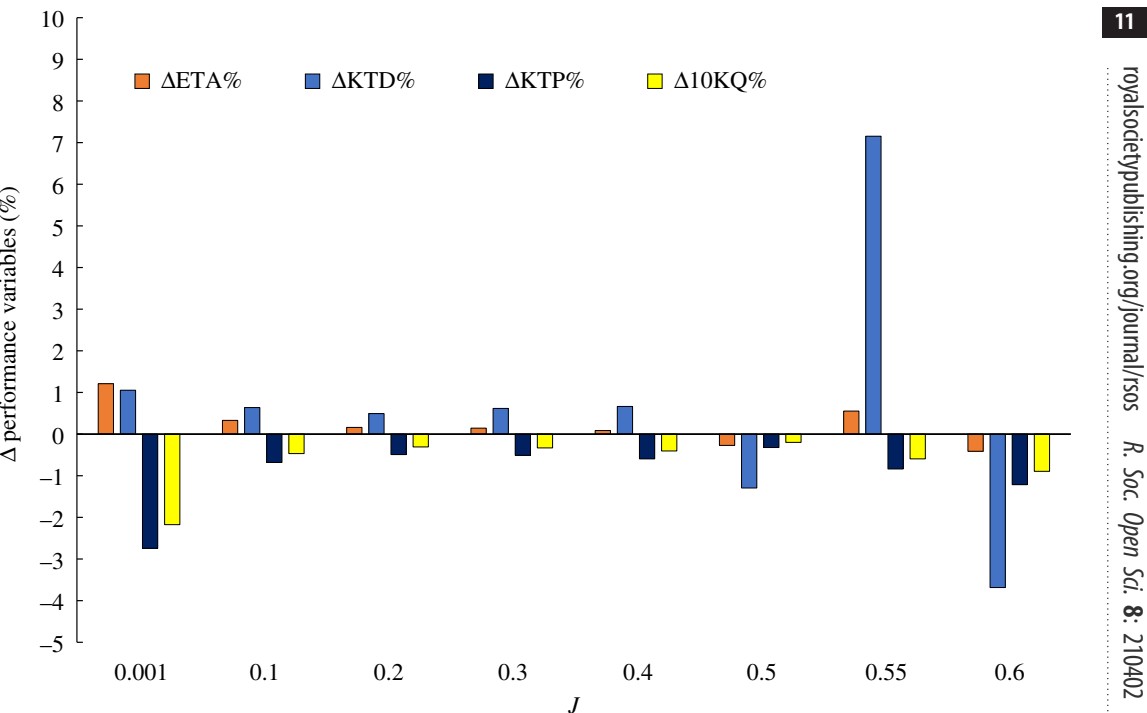

**Figure 9.** Δ% of key performance variables of SLE when compared to REF at a range of J.

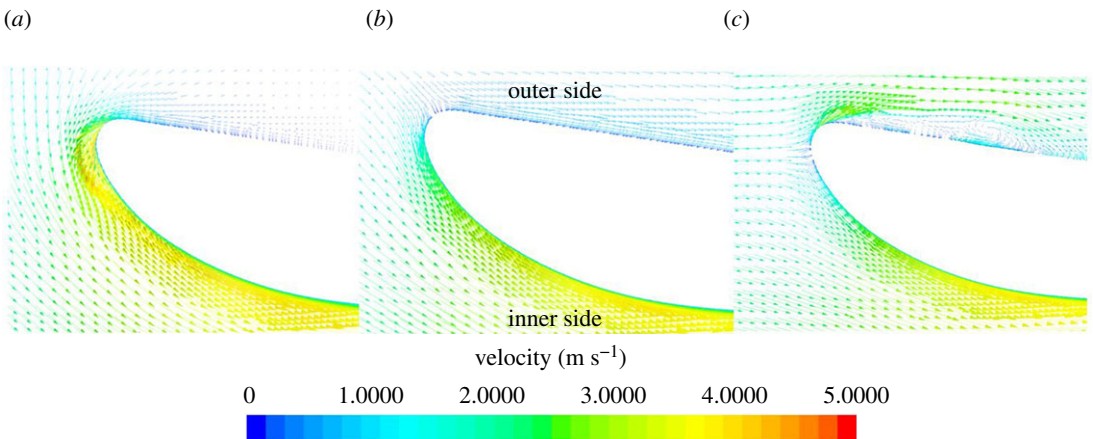

**Figure 10.** Velocity vectors, illustrating inflow angle with varying J, 0.1 (a), 0.3 (b) and 0.55 (c), respectively.

mesh was selected, where a range of operating conditions were considered and are shown in figure 4, with approximately 13 million mesh cells.

## 4.3. Hydrodynamic validation with experimental test

Experimental data acquired by an internal test campaign at CTO, Poland using a KA4–55 and 19A duct were compared with the current numerical results. The description of the geometry can be shown in figure 1 and was replicated in the computational domain. Figure 7 shows the results acquired from the open-water curve characteristics. As can be seen, the computational environment generated can be used to determine the open-water characteristics of the ducted propeller selected in this study to within a good degree of accuracy, with relative errors of total thrust and torque coefficient: 5.1% and 1.4%, respectively.

## 4.4. Validation of the Ffowcs-Williams Hawkings acoustic analogy

In order to validate the FW-H acoustic analogy, the near-field direct hydrodynamic and hydroacoustic pressures were compared for the KA4–55 open propeller at the radial receiver, M0. This has been

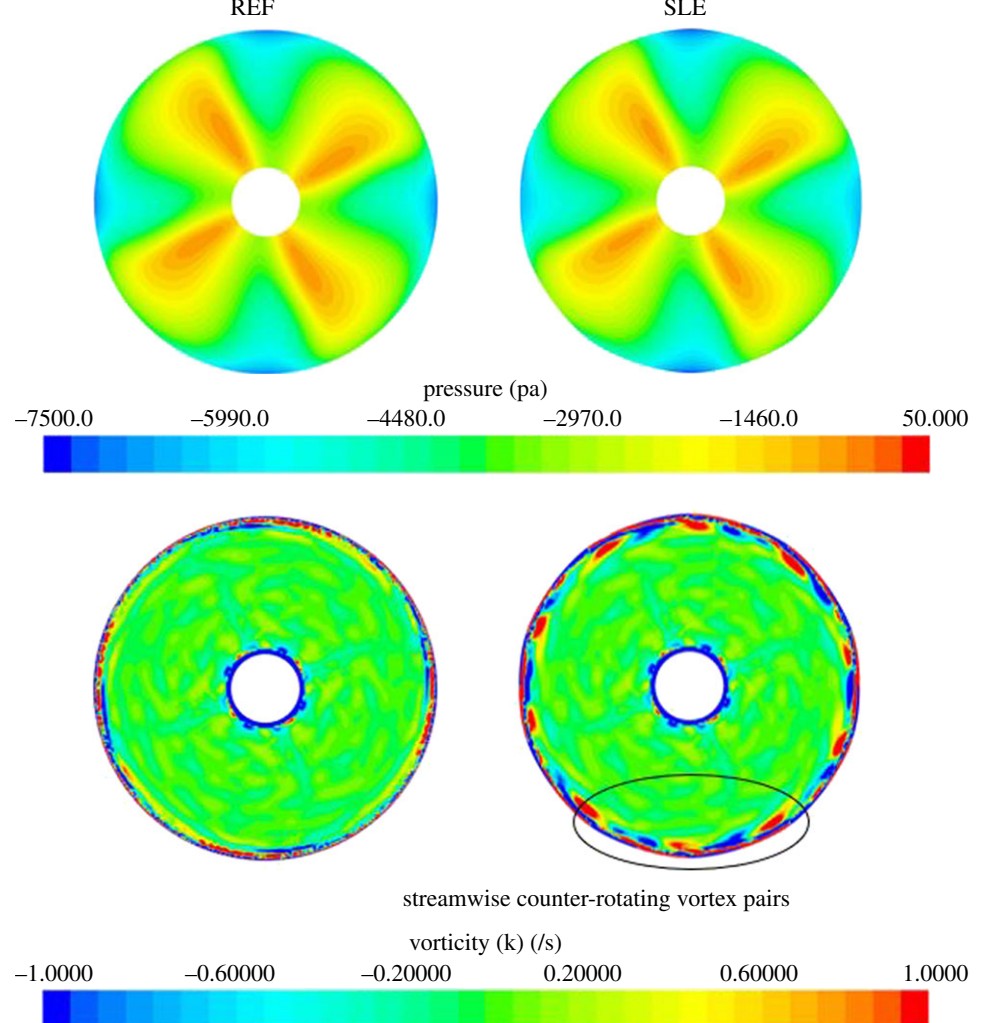

**Figure 11.** Inflow pressure and streamwise vorticity characteristics at $0.26L_{DUCT}$ for both REF and SLE ducts at $J = 0.55$.

conducted in numerous studies within the literature for validation of the FW-H acoustic analogy [4,5,43]. As can be shown in figure 8, there is good agreement between the direct hydrodynamic and hydroacoustic (FW-H) pressure.

# 5. Global hydrodynamic performance results and analysis

Figure 9 shows the percentage difference ($\Delta\%$) of key time-averaged global performance coefficients for the SLE-ducted propeller combination when compared to REF using the sliding mesh technique and DES solver at a range of operating conditions. As can be seen, the duct performance can be enhanced by a maximum of 7.15% at the maximum operating efficiency, increasing the optimum efficiency by roughly 0.55%. It appears that at $J = 0.5$ and 0.6, there is performance degradation due to the inclusion of tubercles. This is likely due to earlier inception of flow separation occurring in the troughs at $J = 0.5$, and the geometrical configuration of tubercles not being prominent enough to manipulate the flow at $J = 0.6$ which will be explained with further analysis. Nonetheless, in between these flow separation conditions, the tubercles can enhance the performance of the duct. Additionally, the LE tubercles result in a reduction of propeller thrust and torque. Therefore, the LE tubercles must influence the inflow characteristics of the propeller.

## 5.1. Propeller inflow characteristics

Figure 10 shows the velocity vector plots for each condition of the REF duct, showing as $J$ is increased, the negative angle of attack is reduced. The duct thrust is created by the difference in the horizontal component of lift and drag which varies with $J$. The pressure and streamwise vorticity inflow

(*a*)

blade thrust versus physical time

(*b*)

amplitude versus frequency

**Figure 12.** Blade force variation versus time (*a,c,e*) and amplitude versus frequency spectrum (*b,d,f*) for REF and SLE at *J* = 0.1, 0.3 and 0.55.

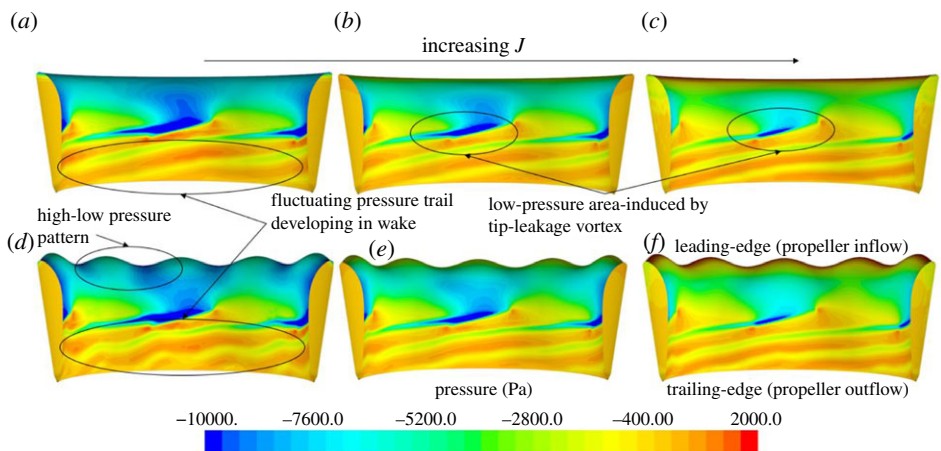

increasing *J*

high-low pressure pattern

fluctuating pressure trail developing in wake

low-pressure area-induced by tip-leakage vortex

leading-edge (propeller inflow)

trailing-edge (propeller outflow)

pressure (Pa)

−10000.    −7600.0    −5200.0    −2800.0    −400.00    2000.0

**Figure 13.** Surface pressure distributions (inner side) of REF and SLE duct at *J* = 0.1 (*a,d*), 0.3 (*b,e*) and 0.55 (*c,f*).

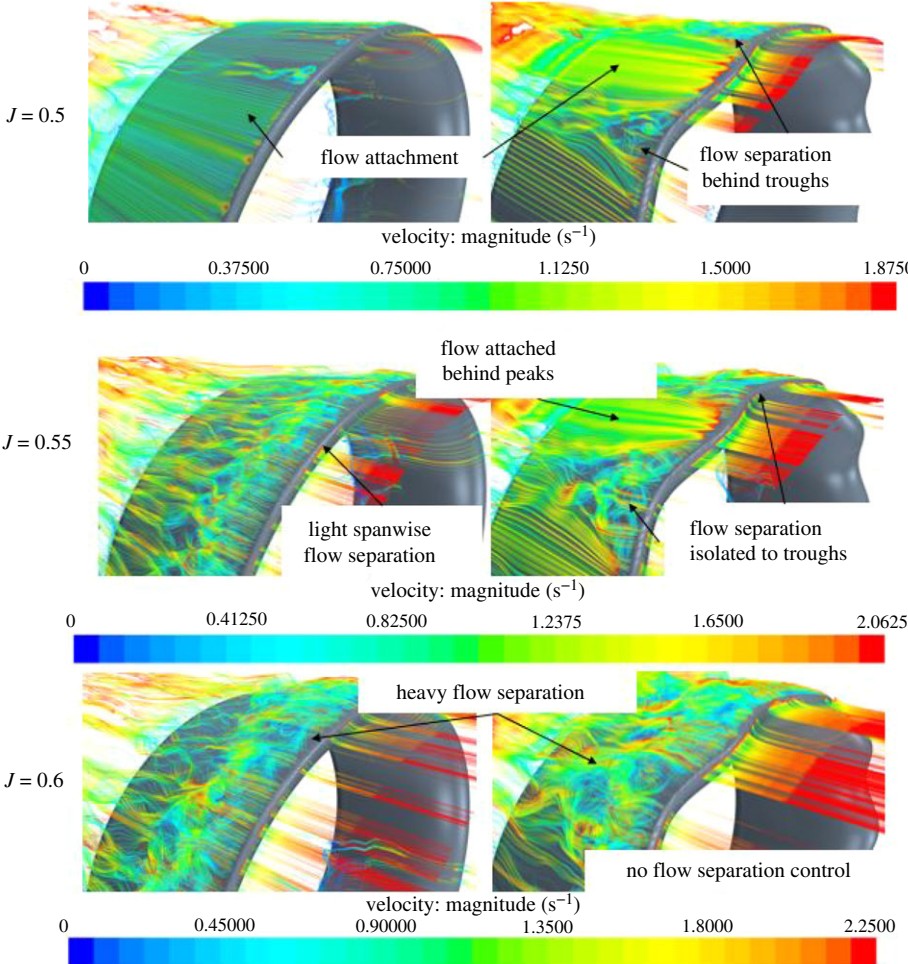

**Figure 14.** Velocity streamlines of REF and SLE duct at $J = 0.5$, 0.55 and 0.6, respectively.

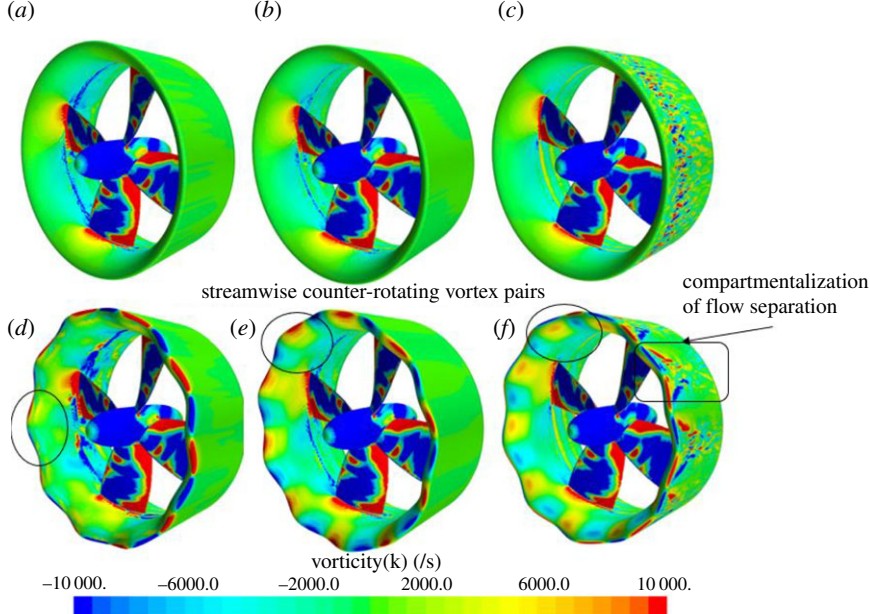

**Figure 15.** Surface streamwise vorticity distributions of REF and SLE duct at $J = 0.1$ ($a,d$), 0.3 ($b,e$) and 0.55 ($c,f$).

characteristics for SLE and REF ducts at $0.26L_{DUCT}$ from the LE of the REF duct or the trough LE section of the SLE duct at an example advance ratio, $J = 0.55$ can be shown in figure 11. As can be seen, the LE tubercles influence the propeller inflow characteristics; this is likely the reason for the variation in blade

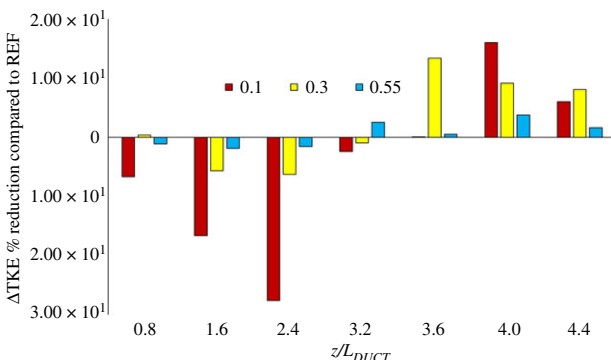

**Figure 16.** TKE reduction (denoted as positive y-axis) of SLE duct compared to REF at several positions in the propeller slipstream for $J = 0.1$, 0.3 and 0.55, respectively.

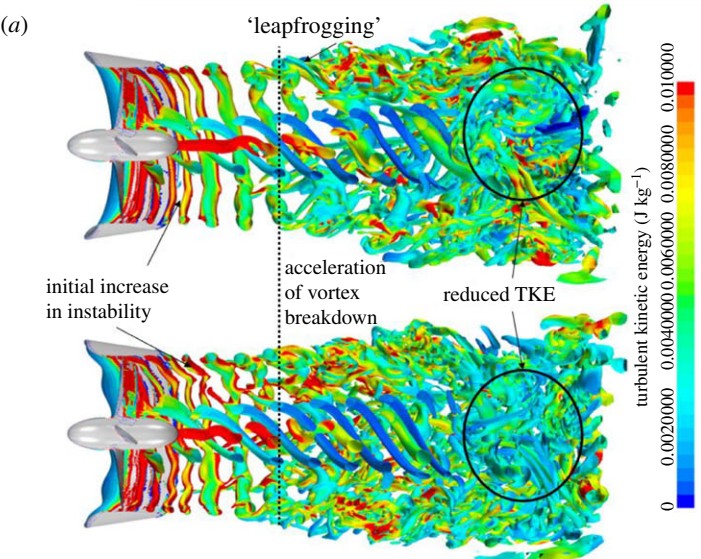

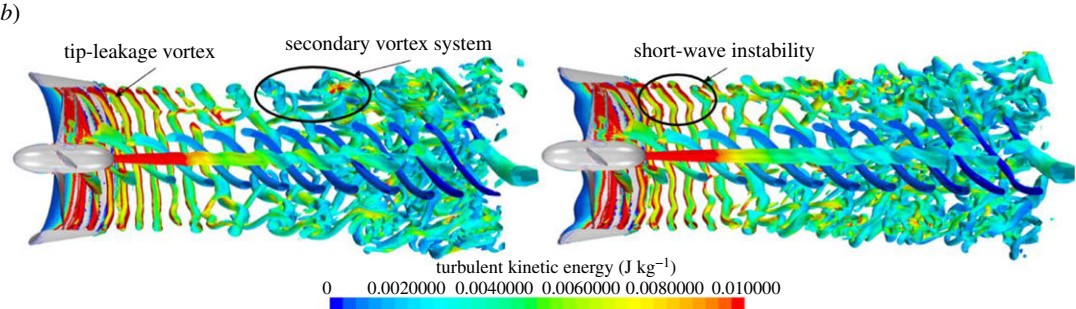

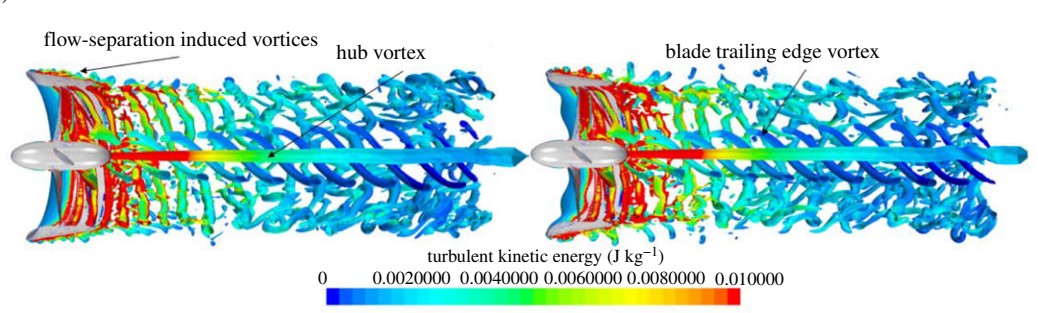

**Figure 17.** Q-criterion plots (right), $\alpha = 1000/s^2$ (coloured by TKE) of REF and SLE ducts at $J = 0.1$, 0.3 and 0.55. (*a*) $J = 0.1$. (*b*) $J = 0.3$. (*c*) $J = 0.55$.

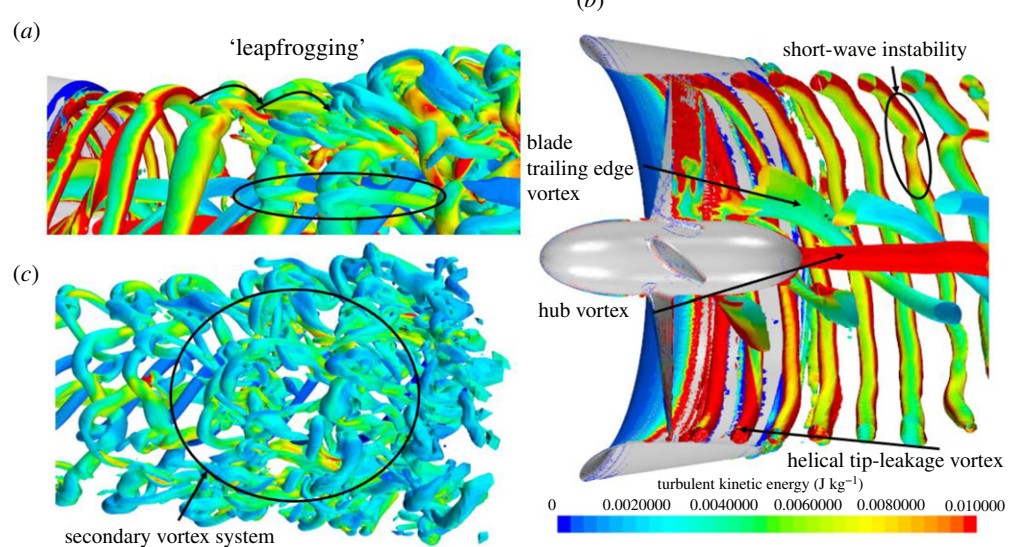

**Figure 18.** Close-up Q-criterion plots (*a–c*), $\alpha = 1000/s^2$ (coloured by TKE) of REF to illustrate fundamental wake vortex dynamics.

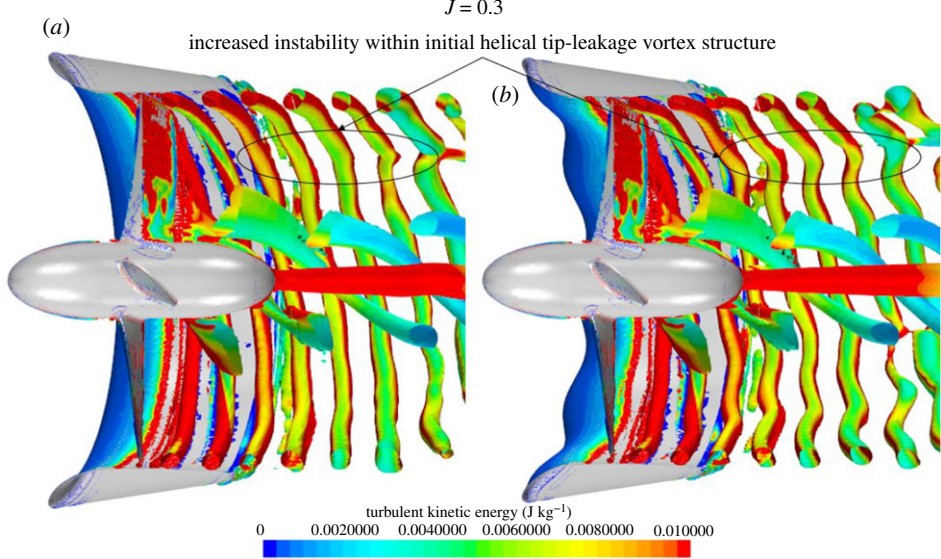

**Figure 19.** Close-up Q-criterion plots (*a*), $\alpha = 1000/s^2$ (coloured by TKE) of REF (*a*) and SLE (*b*) to illustrate the influence of LE tubercles on the helical tip-leakage vortex structure.

thrust and torque. Although there is no appreciable difference in pressure distribution, the tubercles create the contra-rotating streamwise vortices on the inner side of the duct which will interact with the blade surface near the duct wall. The variation in blade thrust and torque with *J* number is likely due to the strength of the streamwise vortices varying because the inflow angle will change with the advance ratio. The blade thrust variation due to the inclusion of the LE tubercles can be further assessed by using FFT to transfer from the time to frequency domain, and this can be shown in the following chapter.

## 6. Blade force results and analysis

The fluctuation of the thrust produced by a single propeller blade is displayed in figure 12*a,c,e* for *J* = 0.1, 0.3 and 0.55. As can be seen, the inclusion of the LE tubercles influences the blade thrust variation over the time period. The blade force variation over time is converted into the frequency domain using FFT analysis and can be shown in figure 12*b,d,f* for *J* = 0.1, 0.3 and 0.55. Distinct peaks can be observed at the

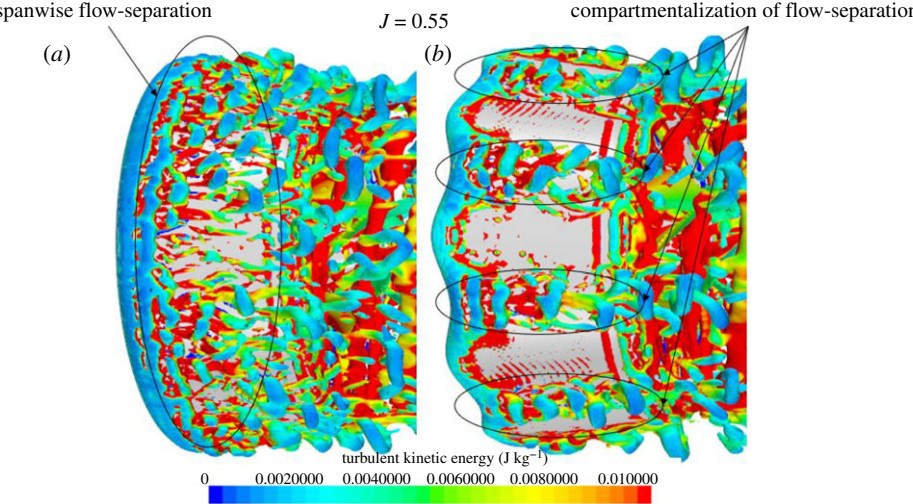

spanwise flow-separation    $J = 0.55$    compartmentalization of flow-separation

(a)    (b)

turbulent kinetic energy (J kg⁻¹)

0    0.0020000    0.0040000    0.0060000    0.0080000    0.010000

**Figure 20.** Close-up Q-criterion plots (a), $\alpha = 1000/s^2$ (coloured by TKE) of REF (a) and SLE (b) to illustrate the influence of LE tubercles on the flow separation-induced vorticity.

first BPF (60 Hz) and two sub-harmonics 15 and 30 Hz for both ducted propeller combinations. However, a distinct peak can also be observed at 150 Hz for the SLE duct; this is due to the inclusion of the 10 tubercles and can be described as the tubercle passage frequency which is the product of the number of tubercles, 10, and the propeller rotation rate, 15 rps. This interaction is likely the reason for the reduction in time-averaged propeller thrust at all operating conditions considered.

# 7. Pressure, velocity and vorticity results and analysis

## 7.1. Surface pressure and velocity distributions

The distribution of pressure on the SLE and REF duct surface (inner side) at $J = 0.1$, 0.3 and 0.55 can be shown in figure 13. On the suction side of the blade tip, a low-pressure area can be observed which is caused by a vortex that is formed by the rotation of the blade close to the duct wall and is known as a tip-leakage vortex. This type of vortex has a pitch that is considerably lower than the pitch of the blade. The high–low-pressure pattern created by the inclusion of the tubercles at the LE can be seen more prominently at the heavier-loaded condition, $J = 0.1$. At this condition, the duct experiences a high negative angle of attack with respect to the incoming flow. Downstream of the blade, a pressure trail can be shown in all configurations which can be described as the development of the tip-leakage vortex into the propeller slipstream. The inclusion of the LE tubercles can be seen to disrupt this vortex, creating fluctuations in the pressure trail, particularly at the lower $J$ ratio.

Figure 14 displays the differences between REF and SLE duct on the outer side of the duct at $J = 0.5$, 0.55 and 0.6 where flow separation is incepted at $J = 0.5$ for the SLE duct. It conditions where no flow separation is experienced ($J < 0.5$), the improvement in duct thrust is due to the additional lift producing surface area of the tubercle and the low-high pressure pattern on the inner side of the duct resulting in a larger net horizontal force, duct thrust. At $J = 0.5$, the flow separation is initiated earlier behind the troughs of the tubercle SLE duct than compared to the REF duct. This results in an increase in drag which would correspond to the degradation in performance at this condition. At $J = 0.55$, the isolation of flow separation can be shown behind the troughs, with flow attachment behind the peaks, compared to spanwise flow separation of the REF duct; this corresponds to a 7.15% improvement in duct thrust at this condition. At $J = 0.6$, the SLE duct does not appear to be able to manipulate the flow at this condition, showing flow separation behind the peaks and troughs, which is due to the geometrical configuration of the tubercle not being prominent enough to control the flow at this condition where the duct experiences the highest advance velocity considered within the investigation.

## 7.2. Surface streamwise vorticity distribution

Figure 15 shows the surface streamwise vorticity displayed on the ducted propeller at operating conditions $J = 0.1$, 0.3 and 0.55. At all operating conditions, counter-rotating streamwise vortices on the inner side of

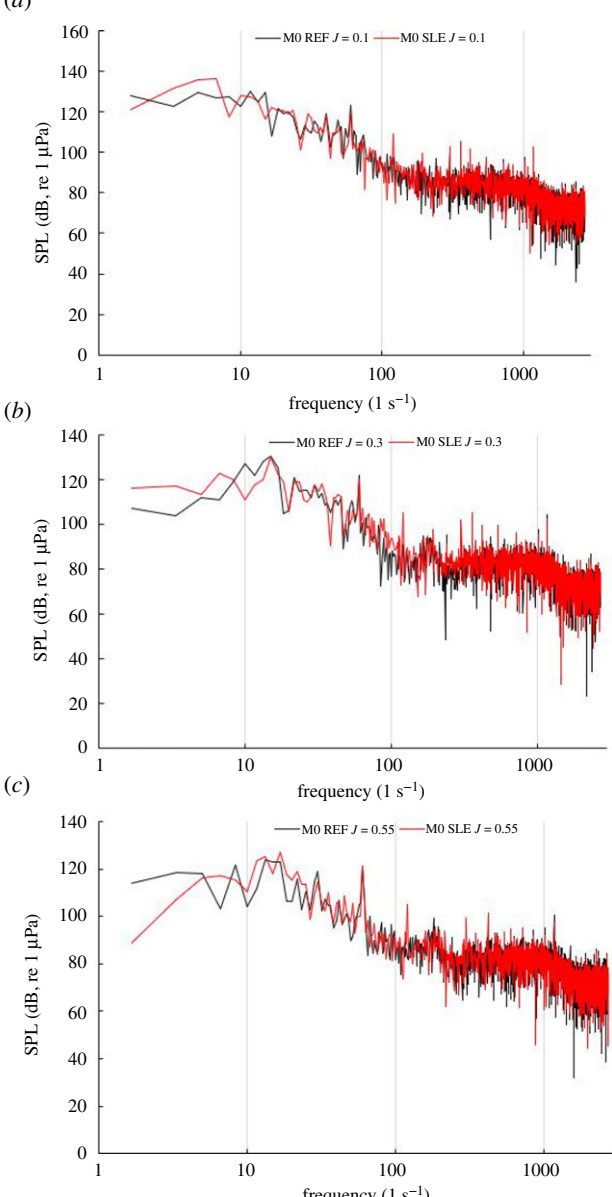

**Figure 21.** Narrowband plot at M0 for REF and SLE combinations at $J = 0.1$ (a). $J = 0.3$ (b) and $J = 0.55$ (c).

the duct can be observed. Where flow separation occurs on the outer side of the duct at $J = 0.55$, a vorticity funnelling pattern can be seen behind the troughs with no vorticity present behind the peaks. This can be compared with the reference design at the same condition, where scattered vorticity can be displayed along the spanwise direction of the duct. Comparing the vortex pair generated at $J = 0.1$ to 0.3 and 0.55, the streamwise vortices are less distinct at the heavier-loaded condition.

## 7.3. Vortex wake dynamics

Turbulent kinetic energy (TKE) surface integrals for seven cross-sections downstream of the ducted propeller, normalized by $L_{DUCT}$, for both REF and SLE were computed. Comparing the SLE to the REF duct, there was a reduction in TKE at the further downstream sections for all operating conditions considered as shown in figure 16. Figure 16 shows the percentage reduction of computed surface integral TKE at several plane sections for the three operating conditions considered, where the positive y-axis denotes a reduction in TKE. The SLE duct initially increases the TKE in the propeller wake but dissipates quicker than the REF duct for all operating conditions further downstream. A maximum of 16% reduction in TKE is observed at $J = 0.1$.

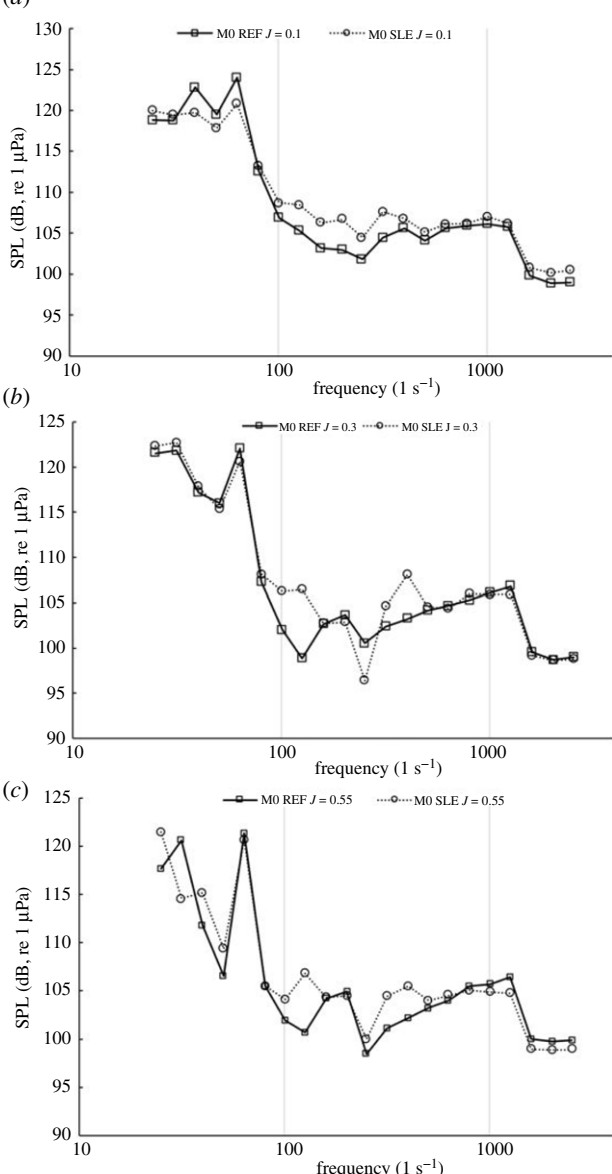

**Figure 22.** Third-octave band plot at M0 for REF and SLE combinations at $J = 0.1$ (a). $J = 0.3$ (b) and $J = 0.55$ (c).

Figure 17 shows the instantaneous half-section Q-criterion plots of the propeller wake coloured by TKE at $J = 0.1$, 0.3 and 0.55 for the REF- and SLE-ducted propeller combinations. The helical vortex structure known as the tip-leakage vortex created by the rotation of the propeller and interaction with the duct wall, blade trailing edge vortex, hub vortex and 'leapfrogging' vortex phenomenon can be observed and are highlighted. The short-wave instability within the helical vortex structure and the secondary vortex system that wraps around the helix can also be shown and highlighted. At the operating condition where the blade loading is the highest, the tip-leakage vortex breaks down a lot quicker downstream when compared to the lower blade loading (higher advance ratio) where the delayed breakdown in the tip-leakage vortex structure can be shown. In addition, at the lower advance ratio, the earlier breakdown of the vortex structure leads to a larger distribution of TKE further downstream. The primary mechanism for the breakdown of the tip-leakage vortex is the short-wave instability, followed by the secondary vortex system. Similar findings were observed by [52] where the stability and breakdown of a helical vortex structure are further explained in [53]. As can be seen, the SLE duct influences the wake dynamics of the ducted propeller when compared to the REF. Immediately at the exit of the ducted propeller, the inclusion of the LE tubercles appears to create additional instability within the helical tip-leakage vortex structure, particularly at $J = 0.1$. This leads to an acceleration in the vortex breakdown into large-scale flow structures and turbulence. Additionally, the reduction in TKE further downstream is particularly apparent at $J = 0.1$. At $J = 0.55$, the flow separation-induced vortex structure can be observed over the outer side of the duct.

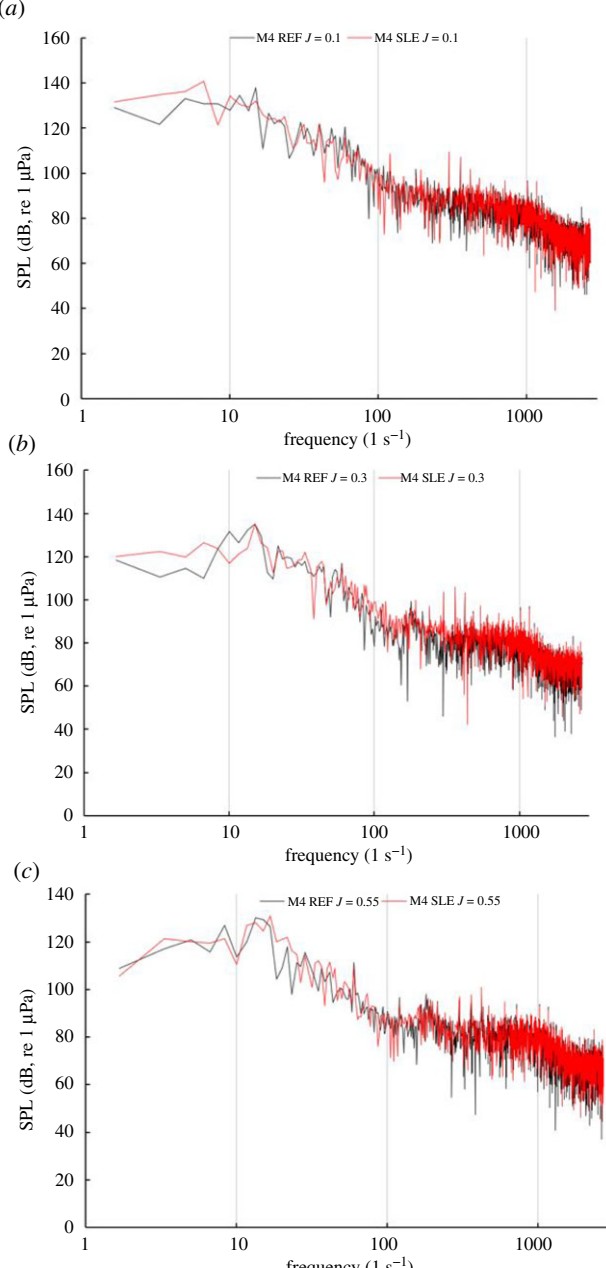

**Figure 23.** Narrowband plot at M4 for REF and SLE combinations at $J = 0.1$ (*a*). $J = 0.3$ (*b*) and $J = 0.55$ (*c*).

The fundamental mechanisms of the wake dynamics of the REF-ducted propeller can be shown clearly in the zoomed-in isometric views depicted in figure 18. Figure 19 shows the close-up views of the influence of the LE tubercles on the short-wave instability within the initial tip-leakage vortex structure. Figure 20 shows the classic tubercle effect, where flow separation occurs on the outer side of the ducted propeller at $J = 0.55$, and the flow separation can be compartmentalized.

# 8. Hydroacoustic performance results and analysis

## 8.1. Near-field noise

Figure 21 shows the narrowband sound spectrum for the REF- and SLE-ducted propeller combinations at the near-field receiver located in the radial position with respect to the propeller plane (M0) at $J = 0.1$, 0.3 and 0.55. At all operating points, the first and second BPFs can be clearly observed by the distinct tonal

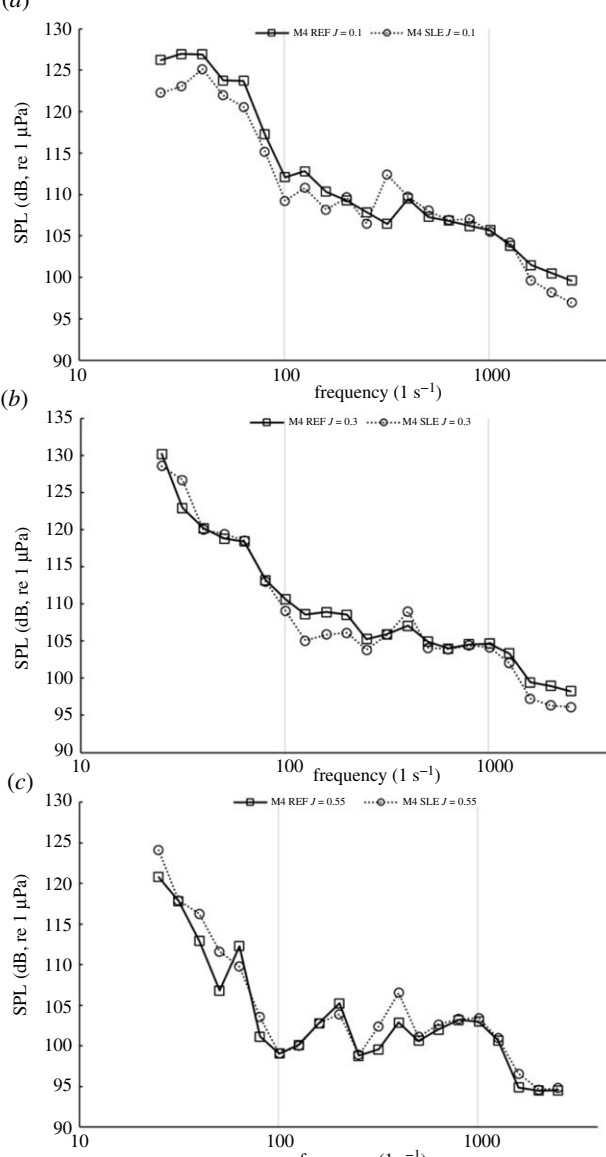

**Figure 24.** Third-octave band plot at M4 for REF and SLE combinations at $J = 0.1$ (a). $J = 0.3$ (b) and $J = 0.55$ (c).

peaks at 60 and 120 Hz (note: BPF = rotation rate (rps) × blade number). Figure 22 shows the third-octave band plots for the same receiver at the same operating conditions enough for flow experienced.

Figure 23 shows the narrowband sound spectrum for the REF- and SLE-ducted propeller combinations at the near-field receiver located at the furthest receiver downstream of the radial position with respect to the propeller plane (M4) at $J = 0.1$, 0.3 and 0.55. In the heavier-loaded conditions, the BPFs, although distinguishable, are not as prominent when comparing to the narrowband spectrum at M0. Although this is the case, the first and second BPF can still be observed at all operating conditions, although they are not as distinct, especially in the heavier-loaded conditions, when compared to spectra at M0. This is likely due to the role of turbulence and vorticity-induced noise, which is more significant at receivers located downstream of the propeller plane than from the radial position (M0) due to being in the vicinity of the propeller slipstream which is where the majority of the turbulent and vorticity-induced noise is generated from. Figure 24 shows the third-octave band plots for the same receiver at the same operating conditions.

Figure 25 shows the SPL levels of the first and second BPF for REF- and SLE-ducted propeller combinations at receivers from M0 to M4 at $J = 0.1$, 0.3 and 0.55. Generally, the first BPF is noisier than the second BPF for all configurations and the SPL of both BPFs reduce the further downstream the receiver is. At $J = 0.1$ and 0.3, the SPL of the first BPF is reduced with the inclusion of the SLE duct when compared to the REF-ducted propeller combination. At $J = 0.55$, the SPL of the first BPF is

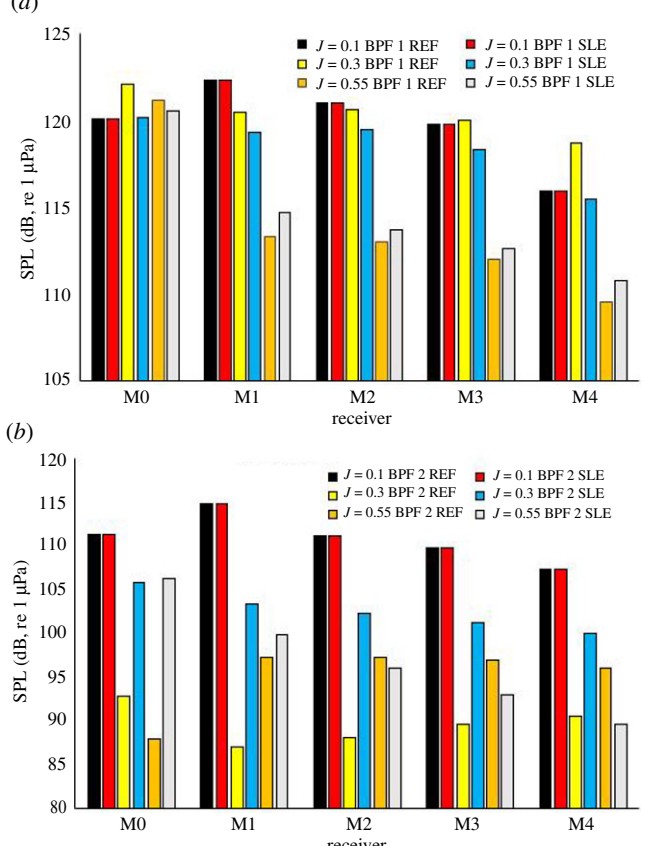

**Figure 25.** First and second BPF SPL levels for REF and SLE ducts at near-field receivers.

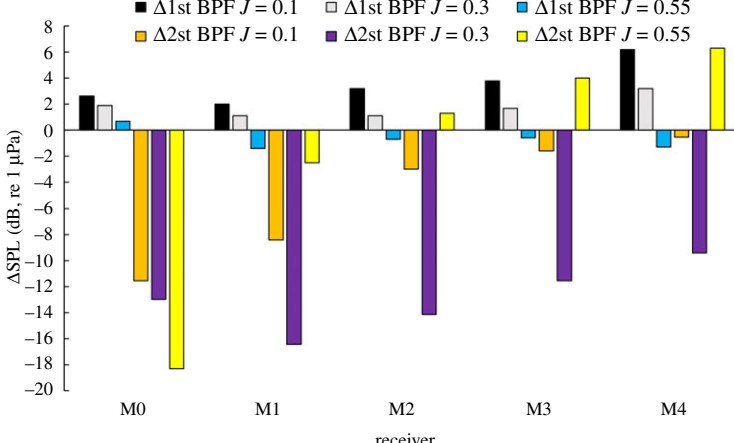

**Figure 26.** Change in first and second BPF of REF- and SLE-ducted propeller combinations at near-field receivers M0–M4 (positive denotes reduction in noise).

increased with the inclusion of the SLE duct when compared to the REF-ducted propeller combination. At M0 and M1, the SPL of the second BPF is reduced when comparing the SLE to the REF duct, but at M2–M4, the SPL increases at $J = 0.3$ and 0.55. The change in first and second BPF at all operating conditions and near-field receivers can be shown in figure 26.

## 8.2. Far-field noise

Figures 27 and 28 shows the narrowband and third-octave band plots for REF and SLE ducts at the radial plane (100D) at operating conditions, $J = 0.1$, 0.3 and 0.55, respectively. As can be shown, the SLE duct

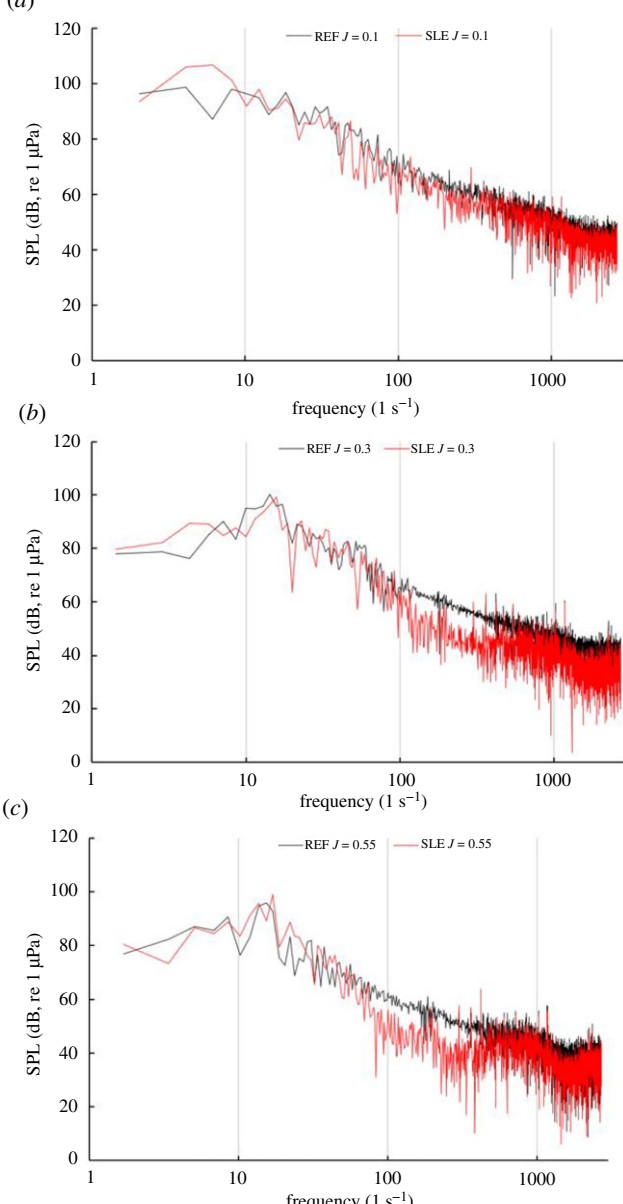

**Figure 27.** Narrowband plots for REF and SLE ducts at radial plane (100D), $J = 0.1$ (a), $J = 0.3$ (b) and $J = 0.55$ (c).

can reduce the noise at all operating conditions when compared to the REF-ducted propeller combination. The change in SPL in the third-octave band can be shown for the same receiver at the same operating conditions in figure 29. The maximum level of reduction at certain frequency ranges is 3.5 dB, 4 dB and 11 dB at $J = 0.1$, 0.3 and 0.55, respectively.

Figure 30 shows the OASPL directivity plots for REF- and SLE-ducted propeller combinations. As can be seen, the SLE duct can reduce the OASPL by a maximum of 3.4 dB at $J = 0.1$, where there is a negligible difference at $J = 0.3$ and an approximate 3 dB increase at $J = 0.55$. The increase in OASPL at the higher $J$ number is likely due to the increase in noise in the low-frequency range (0–100 Hz) which is where the highest levels of SPL occur in the spectrum and thus has the largest weighting when calculating the OASPL.

## 9. Concluding remarks

This study has shown the effect on the hydrodynamic and hydroacoustic performance of the implementation of LE tubercles on the duct of a ducted propeller with detailed flow analysis of the vortex dynamics of the ducted propeller slipstream using a formulation of a DES solver to

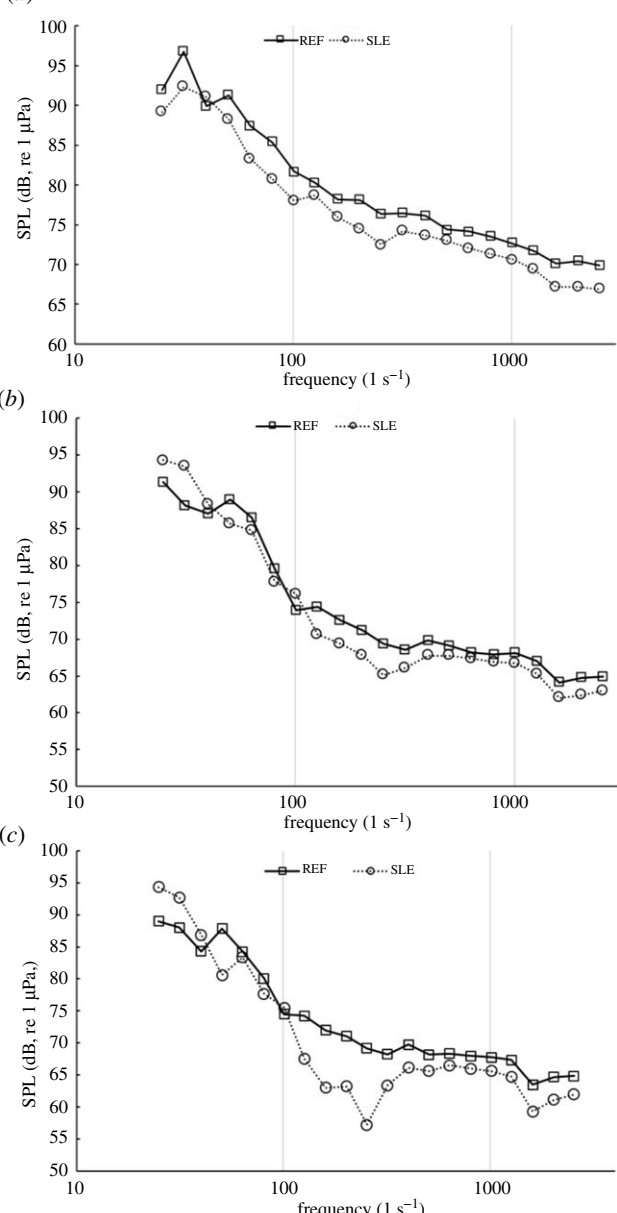

**Figure 28.** Third-octave band plots for REF and SLE ducts at radial plane (100D), $J = 0.1$ (*a*), $J = 0.3$ (*b*) and $J = 0.55$ (*c*).

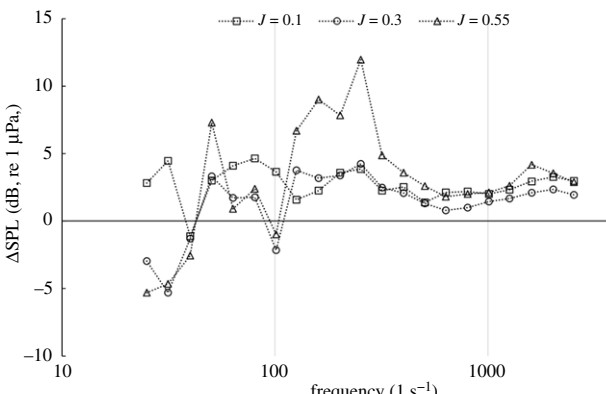

**Figure 29.** OASPL of third-octave band plots for SLE compared to REF duct at a radial plane (100D), $J = 0.1$, 0.3 and 0.55, positive y-axis denotes noise reduction.

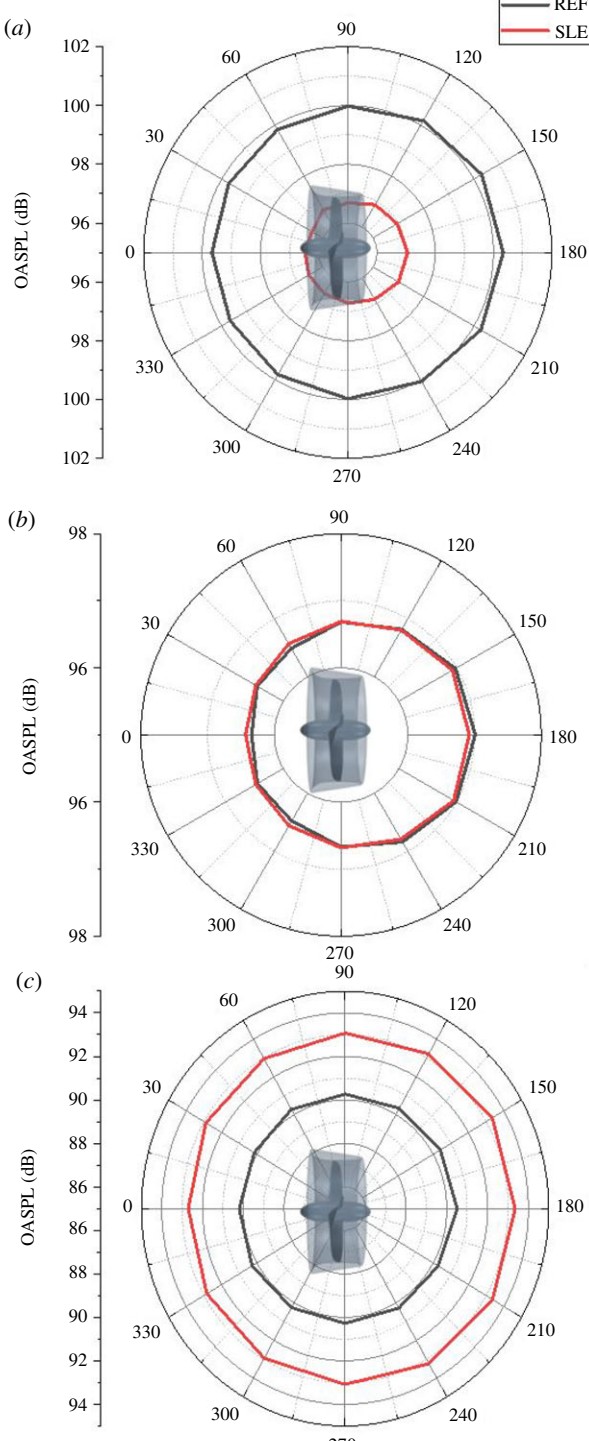

**Figure 30.** OASPL directivity plots for REF and SLE ducts at $J = 0.1$ (*a*), $J = 0.3$ (*b*) and $J = 0.55$ (*c*).

solve the hydrodynamic flow field and the FW-H acoustic analogy to propagate the noise for the first time. The following conclusions can be made from the study:

— LE tubercle-modified duct was investigated where both hydrodynamic and hydroacoustic benefits were observed. Fundamental mechanisms associated with LE tubercles were observed within this study, namely the compartmentalization of flow separation, change in pressure distributions and counter-rotating streamwise vortices.
— The inclusion of LE tubercles will improve the maximum duct thrust capability at most operating conditions considered to a maximum of 7.15% at the maximum efficiency point, $J = 0.55$. At this

condition, the improvement was mainly due to the compartmentalization of flow separation observed on the outer side of the duct.

— At all operating conditions, the inclusion of the SLE duct results in an increase in TKE initially within the slipstream, but then this reduces further downstream to a maximum reduction of 16% at $J = 0.1$.

— At $J = 0.1$, the OASPL was reduced with the inclusion of the SLE duct, where a maximum reduction of 3.4 dB was predicted at $J = 0.1$ and in some frequency ranges in other operating conditions, a maximum of 11 dB reduction was observed. This is largely due to the disruption of the ducted propeller wake structure which will affect the turbulence and vorticity-induced noise.

This study has shown strong evidence to support the application of LE tubercle technology due to its hydrodynamic benefits and noise mitigation capabilities on a benchmark marine-ducted propeller using high-fidelity computational simulations. This could be further supported by a model-scale test campaign to measure the URN from both REF- and SLE-ducted propeller combinations. Although this brings its own challenges, it is a planned future work for the author.

Data accessibility. Models developed and simulated in this paper have now been uploaded to the data repository Dryad for review. Please access through the URL and DOI: doi:10.5061/dryad.6wwpzgmz2. URL: https://doi.org/10.5061/dryad.6wwpzgmz2.

Authors' contributions. C.S. was involved in simulation, research and drafting of the paper. W.S. was involved in modelling, supervision, funding support and review of the paper.

Competing interests. We declare we have no competing interests.

Funding. The funding support from BAE systems plc. is greatly appreciated and acknowledged (Ref: MEIR PhD 16). This study was supported by Royal Society (grant no. RGS\R1\191167).

Acknowledgements. Results were obtained using the ARCHIE-WeSt High-Performance Computer (www.archie-west.ac.uk) based at the University of Strathclyde.

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
