## [Peer Review File · Royal Society Open Science]

Review History

RSOS-210402.R0 (Original submission)

Review form: Reviewer 1

Is the manuscript scientifically sound in its present form?

Yes

Are the interpretations and conclusions justified by the results?

Yes

Is the language acceptable?

Yes

Do you have any ethical concerns with this paper?

No

Have you any concerns about statistical analyses in this paper?

Yes

Recommendation?

Accept with minor revision (please list in comments)

Comments to the Author(s)

This paper applied the leading-edge tubercles of the Humpback whales' pectoral fins to a benchmark ducted propeller, which aims to reduce the noise. This paper is complete and concise, and can draw a lot of attentions in the field of ocean engineering.

The manuscript is deemed ready for publication until some comments are responded and the manuscript should be improved.

1. The author can remove the word of "equation" in the equation numbers.
2. The authors does not provide a sufficient literature review on the researches of leading edge tubercles, thus a lot of important literatures have been missed, such as Fish et al. 1995, Fish et al. 2006, Zhaoyu Wei et al. Ocean Engineering 2015, Zhaoyu Wei et al. AIAA Journal 2018. In addition, some literatures on the noise reduction on the tubercles are missed, the authors should add them in the manuscript.
3. The author should add some discussions on why the tubercles are used on the duct not the blade? And which is better?

Decision letter (RSOS-210402.R0)

Dear Dr Shi:

I am pleased to inform you that your manuscript entitled "Hydroacoustic and hydrodynamic investigation of bio-inspired leading-edge tubercles on marine ducted thrusters" is now accepted for publication in Royal Society Open Science.

Please see the Royal Society Publishing guidance on how you may share your accepted author manuscript at <https://royalsociety.org/journals/ethics-policies/media-embargo/>. After publication, some additional ways to effectively promote your article can also be found here

<https://royalsociety.org/blog/2020/07/promoting-your-latest-paper-and-tracking-your-results/>.

on behalf of Professor Brooke Flammang (Associate Editor) and Professor R. Kerry Rowe (Subject Editor).

Reviewer(s)' Comments to Author:

Reviewer: 1

Comments to the Author(s)

This paper applied the leading-edge tubercles of the Humpback whales' pectoral fins to a benchmark ducted propeller, which aims to reduce the noise. This paper is complete and concise, and can draw a lot of attentions in the field of ocean engineering.

The manuscript is deemed ready for publication until some comments are responded and the manuscript should be improved.

1. The author can remove the word of "equation" in the equation numbers.
2. The authors does not provide a sufficient literature review on the researches of leading edge tubercles, thus a lot of important literatures have been missed, such as Fish et al. 1995, Fish et al. 2006, Zhaoyu Wei et al. Ocean Engineering 2015, Zhaoyu Wei et al. AIAA Journal 2018. In addition, some literatures on the noise reduction on the tubercles are missed, the authors should add them in the manuscript.
3. The author should add some discussions on why the tubercles are used on the duct not the blade? And which is better?
